# ASSESSING ROBUSTNESS VIA SCORE-BASED ADVERSARIAL IMAGE GENERATION

## ABSTRACT

Most adversarial attacks and defenses focus on perturbations within small $\ell_p$-norm constraints. However, $\ell_p$ threat models cannot capture all relevant semantic-preserving perturbations, and hence, the scope of robustness evaluations is limited. In this work, we introduce *Score-Based Adversarial Generation* (ScoreAG), a novel framework that leverages the advancements in score-based generative models to generate adversarial examples beyond $\ell_p$-norm constraints, so-called *unrestricted* adversarial examples, overcoming their limitations. Unlike traditional methods, ScoreAG maintains the core semantics of images while generating realistic adversarial examples, either by transforming existing images or synthesizing new ones entirely from scratch. We further exploit the generative capability of ScoreAG to purify images, empirically enhancing the robustness of classifiers. Our extensive empirical evaluation demonstrates that ScoreAG matches the performance of state-of-the-art attacks and defenses across multiple benchmarks. This work highlights the importance of investigating adversarial examples bounded by semantics rather than $\ell_p$-norm constraints. ScoreAG represents an important step towards more encompassing robustness assessments.

## 1 INTRODUCTION

Ensuring the robustness of machine algorithms against noisy data or malicious interventions has become a major concern in various applications ranging from autonomous driving (Eykholt et al., 2018) and medical diagnostics (Dong et al., 2023) to the financial sector (Fursov et al., 2021). Even though adversarial robustness has received significant research attention (Goodfellow et al., 2014; Madry et al., 2017; Croce & Hein, 2020b), it is still an unsolved problem (Hendrycks et al., 2022).

Most works on adversarial robustness define adversarial perturbations to lie within a restricted $\ell_p$-norm from the input. However, recent works showed that significant semantic changes can occur within common perturbation norms (Tramèr et al., 2020; Gosch et al., 2023), and that many relevant semantics-preserving corruptions lie outside specific norm ball choices. Examples include physical perturbations such as stickers on stop signs (Eykholt et al., 2018) or naturally occurring corruptions such as lighting or fog (Kar et al., 2022; Hendrycks & Dietterich, 2019). Such examples led to the inclusion of a first $\ell_p$-norm independent robustness benchmark to RobustBench (Croce et al., 2020) and a call to further investigation into robustness beyond $\ell_p$-bounded adversaries (Hendrycks et al., 2022). Thus, in this work we address the following research question:

*How can we generate semantic-preserving adversarial examples beyond $\ell_p$-norm constraints?*

We propose to leverage the significant progress in diffusion models (Sohl-Dickstein et al., 2015; Ho et al., 2020) and score-based generative models (Song et al., 2020) in generating realistic images. Specifically, we introduce *Score-Based Adversarial Generation (ScoreAG)*, a framework designed to synthesize adversarial examples, transform existing images into adversarial ones, and purify images. Using diffusion guidance (Dhariwal & Nichol, 2021) ScoreAG can generate semantic-preserving adversarial examples that are not captured by common $\ell_p$-norms (see Fig. 1). Overall ScoreAG represents a novel tool for assessing and enhancing the empirical robustness of image classifiers.

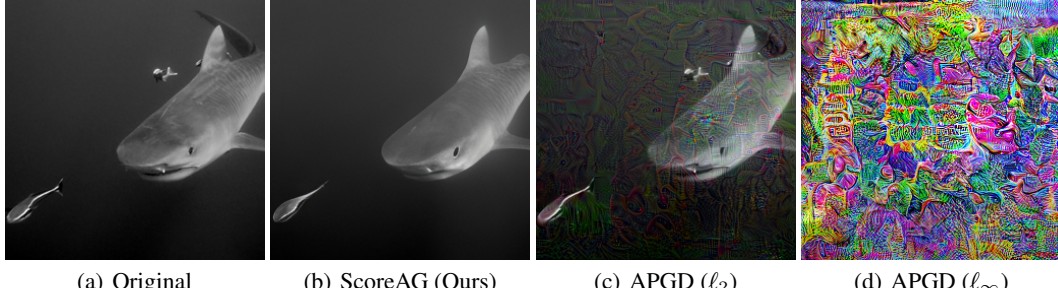

| (a) Original | (b) ScoreAG (Ours) | (c) APGD ($\ell_2$) | (d) APGD ($\ell_\infty$) |

Figure 1: Examples of various adversarial attacks on image of class "tiger shark" (a). Despite the fact that the image generated by ScoreAG (b) lies outside of common $\ell_p$-norm constraints ($\ell_\infty = 188/255$, $\ell_2 = 18.47$), it perfectly preserves image semantics through a realistic perturbation: removing a small fish to change the predicted label to "hammer shark". This is in stark contrast to APGD (Croce & Hein, 2020b) with matching norm constraints, which either (c) results in highly perceptible and unnatural changes, or (d) fails to preserve image semantics completely.

Our key contributions are summarized as follows:

- We overcome limitations of classical $\ell_p$ threat models by proposing ScoreAG, a framework utilizing diffusion guidance on pre-trained models, enabling the generation of *unrestricted* and at the same time semantic-preserving adversarial examples.
- With ScoreAG we *transform* existing images into adversarial ones as well as *synthesize* completely new adversarial examples.
- By leveraging the generative capability of score-based generative models, we show that ScoreAG enhances classifier robustness by *purifying* adversarial perturbations.
- We demonstrate ScoreAG's capability in an exhaustive empirical evaluation and show it is able to compete with existing attacks and defenses on several benchmarks.

## 2 BACKGROUND

**Score-based Generative Modelling.** Score-based generative models (Song et al., 2020) are a class of generative models based on a diffusion process $\{\mathbf{x}_t\}_{t \in [0,1]}$ accompanied by their corresponding probability densities $p_t(\mathbf{x})$. The diffusion process perturbs data $\mathbf{x}_0 \sim p_0$ into a prior distribution $\mathbf{x}_1 \sim p_1$. The transformation can be formalized as a Stochastic Differential Equation (SDE), i.e.,

$$\mathrm{d}\mathbf{x}_t = \mathbf{f}(\mathbf{x}_t, t)\mathrm{d}t + g(t)\mathrm{d}\mathbf{w}, \tag{1}$$

where $\mathbf{f}(\cdot, t): \mathbb{R}^d \to \mathbb{R}^d$ represents the drift coefficient of $\mathbf{x}_t$, $g(\cdot): \mathbb{R} \to \mathbb{R}$ the diffusion coefficient, and $\mathbf{w}$ the standard Wiener process (i.e., Brownian motion). Furthermore, let $p_{st}(\mathbf{x}_t|\mathbf{x}_s)$ describe the transition kernel from $\mathbf{x}_s$ to $\mathbf{x}_t$, where $s < t$. By appropriately choosing $\mathbf{f}$ and $g$, $p_1$ asymptotically converges to an isotropic Gaussian Distribution, i.e., $p_1 \approx \mathcal{N}(\mathbf{0}, \mathbf{I})$. To generate data, the reverse-time SDE needs to be solved:

$$\mathrm{d}\mathbf{x}_t = [\mathbf{f}(\mathbf{x}_t, t) - g(t)^2 \nabla_{\mathbf{x}_t} \log p_t(\mathbf{x}_t)]\mathrm{d}t + g(t)\mathrm{d}\mathbf{w}. \tag{2}$$

Solving the SDE requires the time-dependent score function $\nabla_{\mathbf{x}_t} \log p_t(\mathbf{x}_t)$, which is commonly estimated using a neural network $s_\theta(\mathbf{x}_t, t)$. The parameters of this network can be learned through the following optimization problem:

$$\theta = \arg\min_\theta \mathbb{E}_t \left\{ \lambda(t) \mathbb{E}_{\mathbf{x}_0} \mathbb{E}_{\mathbf{x}_t|\mathbf{x}_0} \left[ \|\mathbf{s}_\theta(\mathbf{x}_t, t) - \nabla_{\mathbf{x}_t} \log p_{0t}(\mathbf{x}_t|\mathbf{x}_0)\|_2^2 \right] \right\}. \tag{3}$$

Here, $\lambda(\cdot): [0, 1] \to \mathbb{R}_{>0}$ serves as a time-dependent weighting parameter, $t$ is uniformly sampled from the interval $[0, 1]$, $\mathbf{x}_0 \sim p_0$, and $\mathbf{x}_t \sim p_{0t}(\mathbf{x}_t|\mathbf{x}_0)$.

**Diffusion Guidance.** To enable conditional generation of unconditionally trained diffusion models, Dhariwal & Nichol (2021) introduce classifier guidance. The central idea of this approach is to leverage Bayes' theorem for the computation of the conditional gradient, enabling the utilization of an unconditional model for conditional tasks and additionally improving the performance of conditional models. This can be formally described as $\nabla_{\mathbf{x}_t} \log p(\mathbf{x}_t|c) = \nabla_{\mathbf{x}_t} \log p(\mathbf{x}_t) + \nabla_{\mathbf{x}_t} \log p(c|\mathbf{x}_t)$, where $\nabla_{\mathbf{x}_t} \log p(c|\mathbf{x}_t)$ denotes the gradient of a classifier and $c$ the guidance condition.

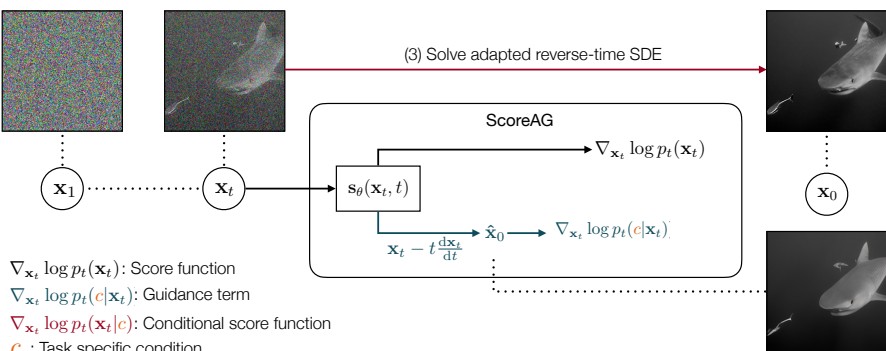

Figure 2: An overview of ScoreAG and its three steps. ScoreAG starts from noise $\mathbf{x}_1$ and iteratively denoises it into an image $\mathbf{x}_0$. It uses the task-specific guidance terms $\nabla_{\mathbf{x}_t} \log p_t(c|\mathbf{x}_t)$ and the score function $\nabla_{\mathbf{x}_t} \log p_t(\mathbf{x}_t)$ to guide the process towards the task specific condition $c$. The network $\mathbf{s}_\theta$ is used for approximating the score function $\nabla_{\mathbf{x}_t} \log p_t(\mathbf{x}_t)$ and for the one-step Euler prediction $\hat{\mathbf{x}}_0$.

## 3 SCORE-BASED ADVERSARIAL GENERATION

In this section we introduce *Score-Based Adversarial Generation* (ScoreAG), a framework employing generative models to evaluate robustness beyond the $\ell_p$-norm constraints. ScoreAG is designed to perform the following three tasks: **(1)** the generation of adversarial images (GAS), **(2)** the transformation of existing images into adversarial examples (GAT), and **(3)** the purification of images to enhance empirical robustness of classifiers (GAP).

ScoreAG consists of three steps: **(1)** select a guidance term for the corresponding task to model the conditional score function $\nabla_{\mathbf{x}_t} \log p(\mathbf{x}_t|c)$, **(2)** adapt the reverse-time SDE with the task-specific conditional score function, and **(3)** solve the adapted reverse-time SDE for an initial noisy image $\mathbf{x}_1 \sim \mathcal{N}(\mathbf{0}, \mathbf{I})$ using numerical methods. Depending on the task, the result is either an adversarial or a purified image. We provide an overview of ScoreAG in Fig. 2.

In detail, the conditional score function is composed of the normal score function $\nabla_{\mathbf{x}_t} \log p_t(\mathbf{x}_t)$ and the task-specific guidance term $\nabla_{\mathbf{x}_t} \log p(c|\mathbf{x}_t)$, that is

$$\nabla_{\mathbf{x}_t} \log p_t(\mathbf{x}_t|c) = \nabla_{\mathbf{x}_t} \log p_t(\mathbf{x}_t) + \nabla_{\mathbf{x}_t} \log p_t(c|\mathbf{x}_t), \tag{4}$$

where $\log p_t(\mathbf{x}_t)$ is modeled by a score-based generative model. Solving the adapted reverse-time SDE yields a sample of the conditional distribution $p(\mathbf{x}_0|c)$, i.e., an adversarial or purified image. To simplify the presentation, we will denote class-conditional functions as $p_y(\mathbf{x}_t)$ rather than the more verbose $p(\mathbf{x}_t|y)$.

Let $y^* \in \{1, \ldots, K\}$ denote the true class of a clean image $\mathbf{x} \in [0, 1]^{C \times H \times W}$, $\tilde{y} \neq y^*$ be a different class, and $f(\cdot) : [0, 1]^{C \times H \times W} \rightarrow \{1, \ldots, K\}$ a classifier. An image $\mathbf{x}_{\text{ADV}} \in [0, 1]^{C \times H \times W}$ is termed an adversarial example if it is misclassified by $f$, i.e., $f(\mathbf{x}) = y^* \neq \tilde{y} = f(\mathbf{x}_{\text{ADV}})$, while preserving the semantics, i.e., $\Omega(\mathbf{x}) = \Omega(\mathbf{x}_{\text{ADV}})$ with $\Omega$ denoting a semantic describing oracle. Therefore, adversarial examples do not change the true label of the image. To enforce this, conventional adversarial attacks restrict the perturbation to lie in a certain $\ell_p$-norm, avoiding large differences to the original image. In contrast, ScoreAG is not limited by $\ell_p$-norm restrictions but preserves the semantics by employing a class-conditional generative model. In the following we introduce each task in detail.

### 3.1 GENERATIVE ADVERSARIAL SYNTHESIS

Generative Adversarial Synthesis (GAS) aims to synthesize images that are adversarial by nature. While these images maintain the semantics of a certain class $y^*$, they are misclassified by a classifier into a different class $\tilde{y}$. The formal objective of GAS is to sample from the distribution $p_{y^*}(\mathbf{x}_0|f(\mathbf{x}_0) = \tilde{y})$, where $f(\mathbf{x}_0) = \tilde{y}$ corresponds to the guidance condition $c$.

Applying Bayes' theorem accordingly to Eq. 4, the conditional score can be expressed as:

$$\nabla_{\mathbf{x}_t} \log p_{t,y^*}(\mathbf{x}_t|f(\mathbf{x}_0) = \tilde{y}) = \nabla_{\mathbf{x}_t} \log p_{t,y^*}(\mathbf{x}_t) + s_y \cdot \nabla_{\mathbf{x}_t} \log p_{t,y^*}(f(\mathbf{x}_0) = \tilde{y}|\mathbf{x}_t), \tag{5}$$

where $s_y$ is a scaling parameter adjusting the strength of the attack. While the term $\nabla_{\mathbf{x}} \log p_{t,y^*}(\mathbf{x}_t)$ can be approximated using a class-conditional neural network $\mathbf{s}_\theta(\mathbf{x}_t, t, y)$, $\nabla_{\mathbf{x}_t} \log p_{t,y^*}(f(\mathbf{x}_0) = \tilde{y}|\mathbf{x}_t)$ requires access to a time-dependent classifier or the original image $\mathbf{x}_0$.

Considering that we are given a pre-trained classifier and cannot retrain it on perturbed images $\mathbf{x}_t$, we propose to approximate $\mathbf{x}_0$ using a one-step Euler method:

$$\hat{\mathbf{x}}_0 = \mathbf{x}_t - t\frac{d\mathbf{x}_t}{dt} \tag{6}$$

That is we approximate $\nabla_{\mathbf{x}_t} \log p_{t,y^*}(f(\mathbf{x}_0) = \tilde{y}|\mathbf{x}_t) \approx \nabla_{\mathbf{x}_t} \log p(f(\hat{\mathbf{x}}_0) = \tilde{y})$, which in practice corresponds to the CELoss of the classifier $f$ for the class $\tilde{y}$ and the input $\hat{\mathbf{x}}_0$. Contrarily to Dhariwal & Nichol (2021), the one-step Euler method provides compatibility with any pre-trained classifier, removing the need for a time-dependently one. Moreover, this can be adapted to discrete-time diffusion models with the approach by Kollovieh et al. (2023). Consequently, ScoreAG can be utilized with pre-trained classifiers and generative models without the need for specific training.

## 3.2 GENERATIVE ADVERSARIAL TRANSFORMATION

While the GAS task synthesized adversarial samples from scratch, Generative Adversarial Transformation (GAT) focuses on transforming existing images into adversarial examples. For a given image $\mathbf{x}^*$ and its corresponding true class label $y^*$, the objective is to sample a perturbed image misclassified into $\tilde{y}$ while preserving the core semantics of $\mathbf{x}^*$. We denote the resulting distribution as $p_{y^*}(\mathbf{x}_0|f(\mathbf{x}_0) = \tilde{y}, \mathbf{x}^*)$, for the guidance condition $c = \{\mathbf{x}^*, f(\mathbf{x}_0) = \tilde{y}\}$ leading to the following conditional score in Eq. 4:

$$\nabla_{\mathbf{x}_t} \log p_{t,y^*}(\mathbf{x}_t|\mathbf{x}^*, f(\mathbf{x}_0) = \tilde{y}) = \nabla_{\mathbf{x}_t} \log p_{t,y^*}(\mathbf{x}_t) + \nabla_{\mathbf{x}_t} \log p_{t,y^*}(\mathbf{x}^*, f(\mathbf{x}_0) = \tilde{y}|\mathbf{x}_t). \tag{7}$$

By assuming independence between $\mathbf{x}^*$ and the adversarial class $\tilde{y}$, we split the guidance term into $s_x \cdot \nabla_{\mathbf{x}_t} \log p_{t,y^*}(\mathbf{x}^*|\mathbf{x}_t) + s_y \cdot \nabla_{\mathbf{x}_t} \log p_{t,y^*}(f(\mathbf{x}_0) = \tilde{y}|\mathbf{x}_t)$, implying that $\tilde{y}$ should not influence the core semantics of the given image. Note that we introduced the two scaling parameters $s_x$ and $s_y$ that control the possible deviation from the original image and the strength of the attack, respectively. While the score function $\nabla_{\mathbf{x}_t} \log p_{t,y^*}(\mathbf{x}_t)$ and the guidance term $\nabla_{\mathbf{x}_t} \log p_{t,y^*}(f(\mathbf{x}_0) = \tilde{y}|\mathbf{x}_t)$ can be modeled as in the GAS setup, we opt for a Gaussian centered at the one-step Euler prediction $\hat{\mathbf{x}}_0$ (Eq. 6) to model the distribution $p_{t,y^*}(\mathbf{x}^*|\mathbf{x}_t)$:

$$p_{t,y^*}(\mathbf{x}^*|\mathbf{x}_t) = \mathcal{N}(\hat{\mathbf{x}}_0, \mathbf{I}). \tag{8}$$

After applying the logarithm, the term simplifies to the mean squared error (MSE). Importantly, this approach allows the sampled image $\mathbf{x}_0$ to approximate $\mathbf{x}^*$ without imposing specific $\ell_p$-norm constraints, marking it as an *unrestricted* attack. Instead, the class-conditional score network $\mathbf{s}_\theta$ ensures that the core semantics are preserved. Therefore, adversarial examples generated by ScoreAG are not completely unrestricted but constrained to the manifold learned by the generative model.

## 3.3 GENERATIVE ADVERSARIAL PURIFICATION

Generative Adversarial Purification (GAP) extends the capability of ScoreAG to counter adversarial attacks. It is designed to purify adversarial images, i.e., remove adversarial perturbations by leveraging the generative capability of the model to enhance the robustness of machine learning models.

Given an adversarial image $\mathbf{x}_{\text{ADV}}$ that was perturbed to induce a misclassification, GAP aims to sample an image from the data distribution that resembles the semantics of $\mathbf{x}_{\text{ADV}}$, which we denote as $p(\mathbf{x}_0|\mathbf{x}_{\text{ADV}})$ with $\mathbf{x}_{\text{ADV}}$ corresponding to the guidance condition $c$. We model its score function analogously to Eq. 7, i.e.,

$$\nabla_{\mathbf{x}_t} \log p_t(\mathbf{x}_t|\mathbf{x}_{\text{ADV}}) = \nabla_{\mathbf{x}_t} \log p_t(\mathbf{x}_t) + s_x \cdot \nabla_{\mathbf{x}_t} \log p_t(\mathbf{x}_{\text{ADV}}|\mathbf{x}_t), \tag{9}$$

where $s_x$ is a scaling parameter controlling the deviation from the input. Note that we omit $y^*$ since there is no known ground-truth class label. As previously, we utilize a time-dependent neural network $\mathbf{s}_\theta$, to approximate the term $\nabla_{\mathbf{x}_t} \log p_t(\mathbf{x}_t)$. The term $p_t(\mathbf{x}_{\text{ADV}}|\mathbf{x}_t)$ is modeled accordingly to Eq. 8, as before assuming it follows a Gaussian distribution with a mean of the one-step Euler prediction $\hat{\mathbf{x}}_0$. Note that ScoreAG, such as other purification methods, cannot detect adversarial images. Therefore, they also need to preserve image semantics if there is no perturbation.

# 4 EXPERIMENTAL EVALUATION

The primary objective of our experimental evaluation is to assess the capability of ScoreAG in generating adversarial examples. More specifically, we investigate the following properties of ScoreAG: **(1)** the ability to synthesize adversarial examples from scratch (GAS), **(2)** the ability to transform existing images into adversarial examples (GAT), and **(3)** the enhancement of classifier robustness by leveraging the generative capability of the model to purify images (GAP). This evaluation aims to provide comprehensive insight into the strengths and limitations of ScoreAG in the realm of adversarial example generation and classifier robustness.

**Baselines.** In our evaluation we benchmark our adversarial attacks against a wide range of established methods covering various threat models. Specifically, we consider the fast gradient sign-based approaches FGSM (Goodfellow et al., 2014), DI-FGSM (Xie et al., 2019), and SI-NI-FGSM (Lin et al., 2019). In addition, we include comparisons with Projected Gradient Descent-based techniques, specifically Adaptive Projected Gradient Descent (APGD) and its targeted variant (APGDT) (Croce & Hein, 2020b). For a comprehensive assessment, we also examine single pixel, black-box, and minimal perturbation methods, represented by OnePixel (Su et al., 2019), Square (Andriushchenko et al., 2020) and Fast Adaptive Boundary (FAB) (Croce & Hein, 2020a), respectively. Finally, we compare to the unrestricted attacks PerceptualPGDAttack (PPGD), FastLagrangePerceptualAttack (LPA) (Laidlaw et al., 2020), and DiffAttack (Chen et al., 2023a), which is based on latent diffusion.

To evaluate the efficacy of ScoreAG in purifying adversarial examples, we conduct several experiments in a preprocessor-blackbox setting. For the evaluation we employ the targeted APGDT and untargeted APGD attacks (Croce & Hein, 2020b) and ScoreAG in the GAS setup. Our experiments also incorporate the purifying methods ADP (Yoon et al., 2021) and DiffPure (Nie et al., 2022). Additionally, we compare with state-of-the-art adversarial training techniques that partially utilize supplementary data from generative models (Cui et al., 2023; Wang et al., 2023; Peng et al., 2023).

**Experimental Setup.** We employ three benchmark datasets for our experiments: Cifar-10, Cifar-100 (Krizhevsky et al., 2009), and TinyImagenet. We utilize pre-trained Elucidating Diffusion Models (EDM) in the variance preserving (VP) setup (Karras et al., 2022; Wang et al., 2023) for image generation. As our classifier, we opt for the well-established WideResNet architecture WRN-28-10 (Zagoruyko & Komodakis, 2016). The classifiers are trained for 400 epochs using SGD with Nesterov momentum of 0.9 and weight decay of $5 \times 10^{-4}$. Additionally, we incorporate a cyclic learning rate scheduler with cosine annealing (Smith & Topin, 2019) with an initial learning rate of 0.2. To further stabilize the training process, we apply exponential moving average with a decay rate of 0.995. Each classifier is trained four times to ensure the reproducibility of our results. For the restricted methods, we consider the common norms in the literature $\ell_2 = 0.5$ for Cifar-10 and Cifar-100, $\ell_2 = 2.5$ for TinyImagenet, and $\ell_\infty = 8/255$ for all three datasets. For DiffAttack and DiffPure we take the implementation of the official repositories, while the use Torchattacks (Kim, 2020) for the remaining baselines.

**Evaluation metrics.** To quantitatively evaluate our results we compute the adversarial accuracy, i.e., the accuracy after an attack, and the robust accuracy, i.e., the accuracy after an attack on a robust model or after a defense. Furthermore, we use the clean accuracy, i.e., the accuracy of a (robust) model without any attack. Finally, we compute the FID score (Heusel et al., 2017) to measure the quality of the synthetic sample in the GAS task.

## 4.1 QUANTITATIVE RESULTS

**Evaluating Generative Adversarial Synthesis.** As explained in Sec. 3.1, ScoreAG is capable of synthesizing adversarial examples. Fig. 4(a) shows the decline in the accuracy and FID score of a WRN-28-10 classifier as $s_y$ increases. Notably, the classifier yields nearly identical performance as on real data when $s_y = 0$. However, even a minor increase of $s_y$ to 0.125 results in a substantial reduction in accuracy while maintaining a low FID score. Setting $s_y$ to 1.0 causes the classifier's performance to drop below random guessing levels for the Cifar-10 dataset. Additionally, Fig. 3(a) presents sample images generated at various scales. Notably, increasing $s_y$ leads to subtle modifications in the images. Rather than introducing random noise, these changes maintain image coherence

Table 1: Adversarial accuracy for various attacks on the three datasets Cifar-10, Cifar-100, and TinyImagenet. Best scores are in **bold**, second best underlined.

| Dataset | Cifar-10 | Cifar-100 | TinyImagenet |
|---|---|---|---|
| **$\ell_\infty$ restricted** | | | |
| FGSM (Goodfellow et al., 2014) | 31.47±13.39 | 10.82±1.62 | 1.42±0.17 |
| DI-FGSM (Xie et al., 2019) | 0.54±0.54 | 0.13±0.10 | 0.04±0.02 |
| SI-NI-FGSM (Lin et al., 2019) | 3.01±0.93 | 1.20±0.16 | 0.69±0.11 |
| APGD (Croce & Hein, 2020b) | 0.18±0.21 | 0.10±0.03 | 0.18±0.03 |
| APGDT (Croce & Hein, 2020b) | **0.00±0.00** | **0.00±0.00** | **0.00±0.00** |
| Square (Andriushchenko et al., 2020) | 0.25±0.24 | 0.19±0.04 | 0.51±0.05 |
| FAB (Croce & Hein, 2020a) | 1.67±1.56 | 0.76±0.06 | 0.11±0.19 |
| **$\ell_2$ restricted** | | | |
| APGD (Croce & Hein, 2020b) | 1.21±0.05 | 0.69±0.01 | 0.15±0.05 |
| APGDT (Croce & Hein, 2020b) | 0.11±0.01 | 0.09±0.01 | **0.00±0.00** |
| Square (Andriushchenko et al., 2020) | 19.67±0.27 | 7.02±0.42 | 1.26±0.10 |
| FAB (Croce & Hein, 2020a) | 7.41±6.19 | 1.44±0.33 | 0.01±0.01 |
| **$\ell_0$ restricted** | | | |
| OnePixel (Su et al., 2019) | 82.82±0.94 | 59.17±0.77 | 59.42±0.38 |
| **Unrestricted** | | | |
| PPGD (Laidlaw et al., 2020) | 31.82±2.77 | 39.76±2.08 | 2.76±0.10 |
| LPA (Laidlaw et al., 2020) | 0.04±0.05 | **0.00±0.00** | **0.00±0.00** |
| DiffAttack (Chen et al., 2023a) | 14.40±0.97 | 4.89±1.57 | 2.13±0.09 |
| ScoreAG (Ours) | 0.10±0.09 | 0.02±0.03 | **0.00±0.00** |

$s_y = 0$  $s_y = 2^{-3}$  $s_y = 2^{-2}$  $s_y = 2^{-1}$  $s_y = 2^0$     Original  $s_x = 32$  $s_x = 48$  $s_x = 64$  $s_x = 96$

(a) Synthesis (GAS).         (b) Transform (GAT).

Figure 3: Examples on the Cifar-10 dataset. Fig. 3(a) shows the synthesis (GAS) setup and generates images of the classes "horse", "truck", and "deer", which are classified as "automobile", "ship", and "horse", respectively, as $s_y$ increases. Fig. 3(b) shows the transformation (GAT) setup and transforms images of the classes "ship", "horse", and "dog", into adversarial examples classified as "ship", "deer", and "cat". For $s_x = 32$, the images are outside of common perturbation norms but preserve image semantics.

up to a scale of $s_y = 0.5$. Beyond this point, specifically at $s_y = 1.0$, there is a noticeable decline in image quality, as reflected by the FID score.

Since our approach leverages a generative model, it enables the synthesis of an unlimited number of adversarial examples, thereby providing a more comprehensive robustness assessment. Moreover, in scenarios requiring the generation of adversarial examples, our method allows for rejection sampling at low $s_y$ scales, ensuring the preservation of image quality. This is particularly important for adversarial training, where synthetic images can enhance robustness (Wang et al., 2023).

**Evaluating Generative Adversarial Transformation.** Beyond the synthesis of new adversarial examples, our framework allows converting pre-existing images into adversarial ones as described in Sec. 3.2. We show the accuracies of various attacks in Tab. 1. Notably, ScoreAG consistently achieves lower accuracies than the $\ell_2$ and $\ell_0$ restricted methods across all three datasets. The only approaches outperforming it are APGDT with the $\ell_\infty$ norm and LPA. This demonstrates ScoreAG's capability of generating adversarial examples. Surprisingly, the other unrestricted diffusion-based method, DiffAttack, yields considerably higher accuracies. We attribute this discrepancy to the fact that it only leverages the last few iterations of the denoising diffusion process.

While most baselines, with the exception of FAB and DiffAttack, only assess the robustness of adversarial examples on the $\ell_p$-constraint border, ScoreAG draws samples from the distribution of semantic-preserving adversarial examples, as explained in Sec. 3.2, resulting in a more comprehensive robustness evaluation. Intuitively, $s_y$ controls the strength of the attack, while $s_x$ determines the possible deviation from the original image.

Table 2: CIFAR-10 robust accuracy of different adversarial training and purification methods for the attacks APGD, APGDT, and ScoreAG. If multiple threat models exist, we denote results as $\ell_\infty/\ell_2$.

| Model | Clean Accuracy | APGD $\ell_\infty$ | APGD $\ell_2$ | APGDT $\ell_\infty$ | APGDT $\ell_2$ | ScoreAG-GAT (Ours) | Architecture |
|---|---|---|---|---|---|---|---|
| **Adversarial Training** | | | | | | | |
| (Cui et al., 2023) | 92.16 | 70.36 | - | 68.43 | - | 81.94 | WRN-28-10 |
| (Wang et al., 2023) | 92.44 / 95.16 | 70.08 | 84.52 | 68.04 | 83.88 | 76.19 / 77.79 | WRN-28-10 |
| (Wang et al., 2023) | 93.25 / 95.54 | 73.29 | 85.65 | 71.42 | 85.28 | 73.67 / 79.47 | WRN-70-16 |
| (Peng et al., 2023) | 93.27 | 73.67 | - | 71.82 | - | 71.92 | RaWRN-70-16 |
| **Adversarial Purification** | | | | | | | |
| ADP (Yoon et al., 2021) | 93.09 | - | - | 85.45 | - | - | WRN-28-10 |
| DiffPure (Nie et al., 2022) | 89.02 | 87.72 | 88.46 | 88.30 | 88.18 | 88.57±0.06 | WRN-28-10 |
| ScoreAG-GAP (Ours) | 93.93± 0.12 | **91.34±0.46** | **92.13±1.41** | **90.25±0.44** | **90.89±0.40** | **90.74±0.67** | WRN-28-10 |

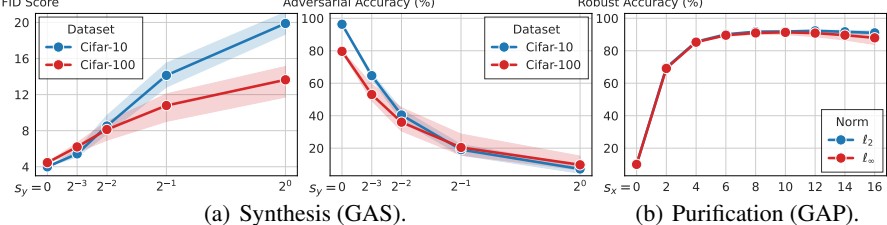

(a) Synthesis (GAS).      (b) Purification (GAP).

Figure 4: The effect of the scale $s_y$ in the synthesis (GAS) setting on the FID score and accuracy, and the effect of the scale $s_x$ in the purification (GAP) setup.

**Evaluating Generative Adversarial Purification.** Finally, we examine the purification ability of ScoreAG. Tab. 2 shows the purification results for various methods on the Cifar-10 dataset.

Our empirical results show that ScoreAG consistently achieves state-of-the-art performance in robust accuracy, outperforming other adversarial purification and training methods. Notably, ScoreAG not only successfully defends attacks but also maintains a high level of clean accuracy comparable to that of adversarial training methods. This demonstrates ScoreAG's capability to preserve the core semantics of images while effectively neutralizing the impact of adversarial perturbations due to their inherently unrealistic nature. Overall, our observations indicate that purification methods can better defend against adversarial attacks than adversarial training approaches, which we attribute to the preprocessor-blackbox setting. Note that it is not possible to detect adversarial examples. Therefore, the purification needs to be applied to all images. However, ScoreAG still achieves a high clean accuracy.

**Hyperparameter study.** We explore the impact of the scale parameters $s_y$ and $s_x$ on accuracy and image purification, as depicted in Fig. 4. In Fig. 4(b), we examine the efficacy of image purification against adversarial attacks of APGD under both $\ell_2$ and $\ell_\infty$ norms across different $s_x$ scales. At $s_x = 0$, the generated images are from an unconditional model without guidance and are independent of the input. Therefore, the robust accuracy equals random guessing. As $s_x$ increases, the accuracy improves, reaching a performance plateau at approximately $s_x = 10$. Increasing $s_x$ further reduces the accuracy as the sampled images start to resemble adversarial perturbations.

Table 3: Adversarial accuracy and median $\ell_2$ distances for various hyperparameter configurations.

| | Adversarial Accuracy in % (↓) | | Median $\ell_2$ distance | |
|---|---|---|---|---|
| Dataset | Cifar-10 | Cifar-100 | Cifar-10 | Cifar-100 |
| **$s_y = 48$** | | | | |
| $s_x = 16$ | **0.10** | **0.02** | 1.12 | 1.09 |
| $s_x = 32$ | 0.23 | **0.02** | 0.65 | 0.64 |
| $s_x = 48$ | 0.32 | 0.03 | 0.49 | 0.49 |
| $s_x = 64$ | 0.34 | 0.04 | 0.43 | 0.40 |
| **$s_y = 64$** | | | | |
| $s_x = 48$ | 0.22 | 0.17 | 0.50 | 0.49 |
| $s_x = 64$ | 0.24 | **0.02** | 0.43 | 0.40 |
| $s_x = 96$ | 0.28 | 0.03 | 0.35 | 0.30 |
| **$s_y = 96$** | | | | |
| $s_x = 48$ | **0.10** | 0.17 | 0.51 | 0.50 |
| $s_x = 64$ | 0.11 | 0.21 | 0.44 | 0.40 |
| $s_x = 96$ | 0.13 | 0.34 | 0.35 | 0.30 |

Finally, Tab. 3 presents the adversarial accuracy and median $\ell_2$ distances across different scale configurations for the Cifar-10 and Cifar-100 datasets. We can observe that an increase in $s_y$ leads to reduced classifier accuracy for Cifar-10, effectively improving the efficacy of the adversarial attacks. A rise in $s_x$, however, increases the accuracy as the generated image closer resembles the original. The median $\ell_2$ distance exhibits a similar behavior. While a lower $s_y$ yields no difference for both datasets, increasing $s_x$ decreases the median distances for Cifar-10 and Cifar-100. In Fig. 3(b), we show examples across various $s_x$ scales on the Cifar-10 dataset. Notably, all scales preserve the image semantics and do not display any observable differences.

## 4.2 QUALITATIVE ANALYSIS

To investigate the quality of the adversarial attacks, we deploy ScoreAG on the ImageNet dataset (Deng et al., 2009) with a resolution of $256 \times 256$. We use the latent diffusion model DiT proposed by Peebles & Xie (2022), along with a pre-trained latent classifier from (Kim et al., 2022). The images are sampled using the denoising procedure by Kollovieh et al. (2023) as explained in Sec. 3.1. Note that as the generative process is performed in the latent space, the model has more freedom in terms of reconstruction.

We show an example image of a tiger shark in Fig. 1 with corresponding adversarial attacks. While the classifier correctly identifies the tiger shark in the baseline image, it fails to do so in the generated adversarial examples. Notably, the $\ell_p$-bounded methods display noticeable noisy fragments. In contrast, ScoreAG produces clean adversarial examples, altering only minor details while retaining the core semantics — most notably, the removal of a small fish — which prove to be important classification cues. We provide further examples for GAS in Sec. B.4 and for GAT in Sec. B.5. The synthetic images display a high degree of realism and the transformed images show visible differences while preserving the semantics of the original image.

## 4.3 HUMAN STUDY

To evaluate whether ScoreAG generates semantic-preserving adversarial examples, we perform a human study on adversarially modified (real) as well as synthetically generated Cifar-10 images. In particular, we sample five images at random from each class to generate 50 adversarial examples using $s_x = 16$ and $s_y = 48$. These adversarial examples have an average $\ell_2$-norm difference to their clean counterparts of $0.68 \pm 0.24$, exceeding the common $\ell_2$-norm ball constraint of $0.5$ (Croce et al., 2020) by on average 36%. For the synthetic examples, we generate 50 images without ($s_y = 0$) and 50 images with guidance

Table 4: Human study to evaluate the adversarial examples of ScoreAG. The human ACC corresponds to the majority vote over all evaluators. Krippendorff's alpha ($\alpha$) indicates the agreement between human evaluators.

| Dataset | Model ACC | Human ACC | $\alpha$ |
|---|---|---|---|
| **Clean** | | | |
| Real | 98% | 100% | 0.854 |
| Synthetic | 94% | 94% | 0.705 |
| **Adversarial** | | | |
| Real | 2% | 94% | 0.792 |
| Synthetic | 0% | 70% | 0.479 |

($s_y = 0.125$), again in a class-balanced fashion. For the adversarial guided synthetic examples, we employ rejection sampling to only consider images that lead to misclassification by the classifier. To ensure high data quality for the study, we used the Prolific platform (Eyal et al., 2021) to employ 60 randomly chosen human evaluators to label the 200 images. To avoid bias, we presented the adversarial examples (synthetic or modified) before the unperturbed examples and introduced the additional category "Other / I don't know".

We compute human accuracy by choosing the majority vote class of all 60 human evaluators and compare it with the ground truth class. We show the results of the human study in Tab. 4. While the model fails to correctly classify the adversarial examples, the human accuracy is still significantly high (94% for real and 70% for synthetic adversarial images) indicating *strong semantic preservation*. Notably, humans can still accurately classify *nearly all* of the adversarial modified images despite significantly larger $\ell_2$ distances. The accuracy is lower but still high for the synthetic images. The drop in accuracy can be in parts explained by the low resolution of CIFAR-10 images - making some (also real) examples difficult to visually classify. This is also captured by Krippendorff's alpha (Krippendorff, 2018), which we use to study the agreement between human evaluators. It indicates whether the agreement is systematic (with $\alpha = 1$ for perfect agreement) or purely by chance ($\alpha = 0$). In our study, we achieve $\alpha$-values significantly larger than 0, concluding that the agreement between human evaluators *is systematic* for all datasets.

## 5 RELATED WORK

**Diffusion and Score-Based Generative Models.** Diffusion models (Sohl-Dickstein et al., 2015; Ho et al., 2020) and score-based generative models (Song et al., 2020) received significant attention in recent years, owing to their remarkable performance across various domains (Kong et al., 2020; Lienen et al., 2023; Kollovieh et al., 2023) and have since emerged as the go-to methodologies for

many generative tasks. Dhariwal & Nichol (2021) proposed diffusion guidance to perform conditional sampling using unconditional diffusion models, which has subsequently been extended for more advanced guidance techniques (Nichol et al., 2021; Avrahami et al., 2022). A recent study has shown that classifiers can enhance their robust accuracy when training on images generated by diffusion models (Wang et al., 2023), demonstrating the usefulness and potential of diffusion models in the robustness domain.

**Adversarial Attacks.** An important line of work are white-box approaches, which have full access to the model parameters and gradients, such as the fast gradient sign method (FGSM) introduced by Goodfellow et al. (2014). While FGSM and its subsequent extensions (Xie et al., 2019; Dong et al., 2018; Lin et al., 2019; Wang, 2021) primarily focus on perturbations constrained by the $\ell_\infty$ norm, other white-box techniques employ projected gradient descent and explore a broader range of perturbation norms (Madry et al., 2017; Zhang et al., 2019).

In contrast, black-box attacks are closer to real-world scenarios and do not have access to model parameters or gradients (Narodytska & Kasiviswanathan, 2016; Brendel et al., 2017; Andriushchenko et al., 2020). Two recent works by Chen et al. (2023a) and Xue et al. (2023) propose DiffAttack and Diff-PGD, respectively. Diff-PGD performs projected gradient descent in the latent diffusion space to obtain $\ell_\infty$-bounded adversarial examples, whereas DiffAttack generates *unrestricted* adversarial examples by leveraging a latent diffusion model. However, as both methods employ only the final denoising stages of the diffusion process in a similar fashion to SDEdit (Meng et al., 2021), the adversarial perturbations only incorporate changes of high-level features. Finally, Chen et al. (2023c) optimize the latent space of Stable Diffusion while a concurrent work by Chen et al. (2023b) leverages GradCAM and PGD to use diffusion models for the generation of adversarial attacks.

**Adversarial Purification.** In response to the introduction of adversarial attacks, a variety of adversarial purification methods to defend machine learning models have emerged. Early works by Song et al. (2017) and Samangouei et al. (2018) utilized Generative Adversarial Networks (GANs) to remove adversarial perturbations from images. Following their work, Hill et al. (2020) proposed to use Energy-Based Models (EBMs) coupled with Markov Chain Monte Carlo (MCMC) sampling for adversarial purification. More recent methods have shifted focus towards score-based generative models, like APD (Yoon et al., 2021), and diffusion models, such as DiffPure (Nie et al., 2022). However, it only utilizes the final stages of the denoising process for purification and is thereby limited to only correcting high-level adversarial features.

## 6    DISCUSSION

**Limitations and Future Work.** Our work demonstrates the potential and capabilities of score-based generative models in the realm of adversarial attacks and robustness. While ScoreAG is able to generate and purify adversarial attacks, some considerable drawbacks remain. Primarily, the evaluations remain a challenge and currently require human studies due to the generated *unrestricted* attacks. Moreover, the proposed purifying approach is only applicable to a preprocessor-blackbox setting, as we are not able to compute the gradients of the generative process in an efficient manner. Lastly, integrating certifiable robustness is an open challenge due to the stochastic nature of the generative process and the challenging unbounded threat model. We are hopeful that these drawbacks will be addressed in future work. For instance, the introduction of a new metric to quantify the semantic similarity between the input and the generated adversarial images. To address the gradient computation, one could explore approximations of the gradients of the sampling process.

**Conclusion.** In this work, we address the question of how to generate *unrestricted* adversarial examples. We introduce ScoreAG, a novel framework that bridges the gap between adversarial attacks and score-based generative models. Utilizing diffusion guidance and pre-trained models, ScoreAG can synthesize new adversarial attacks, transform existing images into adversarial examples, and purify images, thereby enhancing the empirical robust accuracy of classifiers. Our results indicate that ScoreAG can effectively generate semantic-preserving adversarial images beyond the limitations of the $\ell_p$-norms. Our experimental evaluation demonstrates that ScoreAG matches the performance of existing state-of-the-art attacks and defenses. We see unrestricted adversarial examples - as generated by our work - as vital to achieve a holistic view of robustness and complementary to hand-picked common corruptions (Kar et al., 2022) or classical $\ell_p$ threat models.

## REPRODUCIBILITY

Our models are implemented using PyTorch with the pre-trained EDM models by Karras et al. (2022) and Wang et al. (2023), and the guidance scores are computed using automatic differentiation. In Tab. 5 and Tab. 6, we give an overview of the hyperparameters of ScoreAG. For the methods DiffAttack, DiffPure, PPGD, and LPA, we use the corresponding authors' official implementations with the suggested hyperparameters. For the remaining attacks, we use Adversarial-Attacks-PyTorch with its default parameters (Kim, 2020).

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

## A HYPERPARAMETERS

In Tab. 5 and Tab. 6, we show an overview of the hyperparameters used to train the classifiers and to evaluate ScoreAG.

| Hyperparameter | Value |
|---|---|
| Number of epochs | 400 |
| Optimizer | SGD |
| Nesterov momentum | 0.9 |
| Weight decay | $5 \times 10^{-4}$ |
| Exponential moving average | 0.995 |
| Learning rate scheduler | Cyclic with cosine annealing |
| Initial learning rate | 0.2 |

Table 5: Hyperparameters used to train the WRN-28-10 classifiers.

| Hyperparameter | Value |
|---|---|
| **Cifar-10** | |
| $s_y$ (GAT) | 96 |
| $s_x$ (GAT) | 48 |
| $s_x$ (GAP) | 10 |
| **Cifar-100** | |
| $s_y$ (GAT) | 64 |
| $s_x$ (GAT) | 64 |
| **TinyImagenet** | |
| $s_y$ (GAT) | 64 |
| $s_x$ (GAT) | 16 |

Table 6: Hyperparameters used to to evaluate ScoreAG.

## B ADDITIONAL RESULTS

### B.1 RUNTIME COMPARISON OF THE ATTACKS.

In Tab. 7, we compare the runtimes in seconds of various methods. The numbers display the average time to generate 16 adversarial examples on the Cifar-10 dataset on a GTX 1080Ti. DiffAttack was the only method with out-of-memory issues (OOM).

| FGSM | DIFGSM | SINIFGSM | Square | FAB | APGD | APGDT | OnePixel | LPA | PPGD | DiffAttack | ScoreAG |
|---|---|---|---|---|---|---|---|---|---|---|---|
| 0.01 | 0.27 | 1.73 | 67.43 | 7.03 | 0.44 | 0.48 | 2.62 | 41.24 | 2.06 | OOM | 246.06 |

Table 7: Average runtimes in seconds of the different attacks on a GTX 1080Ti to generate 16 adversarial images for the dataset Cifar-10.

### B.2 MORE CLASSIFIERS FOR ADVERSARIAL ATTACKS USING GAT

To verify the efficacy of ScoreAG, we evaluate the accuracy of GAT on four more classifiers for the datasets Cifar-10 and Cifar-100 using the same hyperparameters as for the WRN-28-10 architecture. We show the adversarial accuracy in Tab. 8. As we can observe, ScoreAG successfully generates adversarial attacks on various classifiers, reaching accuracies close to 0%.

### B.3 LARGE PERTURBATION NORMS FOR RESTRICTED ADVERSARIAL ATTACKS

In Fig. 5, we show adversarial examples of different attacks for the image in Fig. 1. We use the same distances ScoreAG achieves.

Table 8: Adversarial accuracy of ScoreAG for various classifiers on the datasets Cifar-10 and Cifar-100.

| Classifier | Cifar-10 | Cifar-100 |
|---|---|---|
| ResNet-20 | 0.01 | 0.10 |
| ResNet-56 | 0.03 | 0.13 |
| VGG-19 | 0.52 | 0.26 |
| RepVGG-A2 | 0.04 | 1.94 |

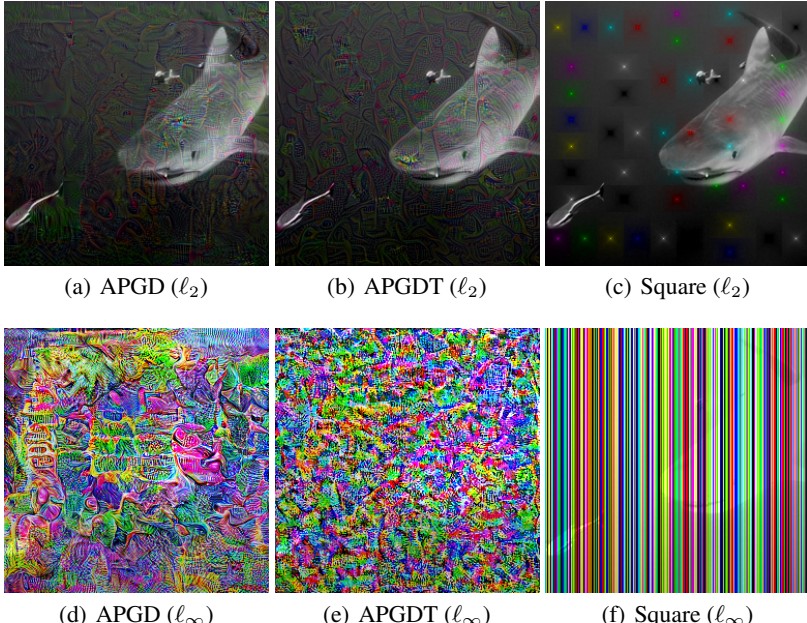

(a) APGD ($\ell_2$)    (b) APGDT ($\ell_2$)    (c) Square ($\ell_2$)

(d) APGD ($\ell_\infty$)    (e) APGDT ($\ell_\infty$)    (f) Square ($\ell_\infty$)

Figure 5: Different adversarial attacks for the example in Fig. 1. The $\ell_\infty$ and $\ell_2$ distances are 188/255 and 18.47, respectively. All methods display major changes in the images compared to the original.

## B.4 GENERATIVE ADVERSARIAL SYNTHESIS

In Fig. 6, we provide additional examples of the GAS task. The images are synthetic adversarial samples of the ImageNet class "indigo bunting". While all images are classified wrongly, most of them contain the right core-semantics and display a high degree of realism.

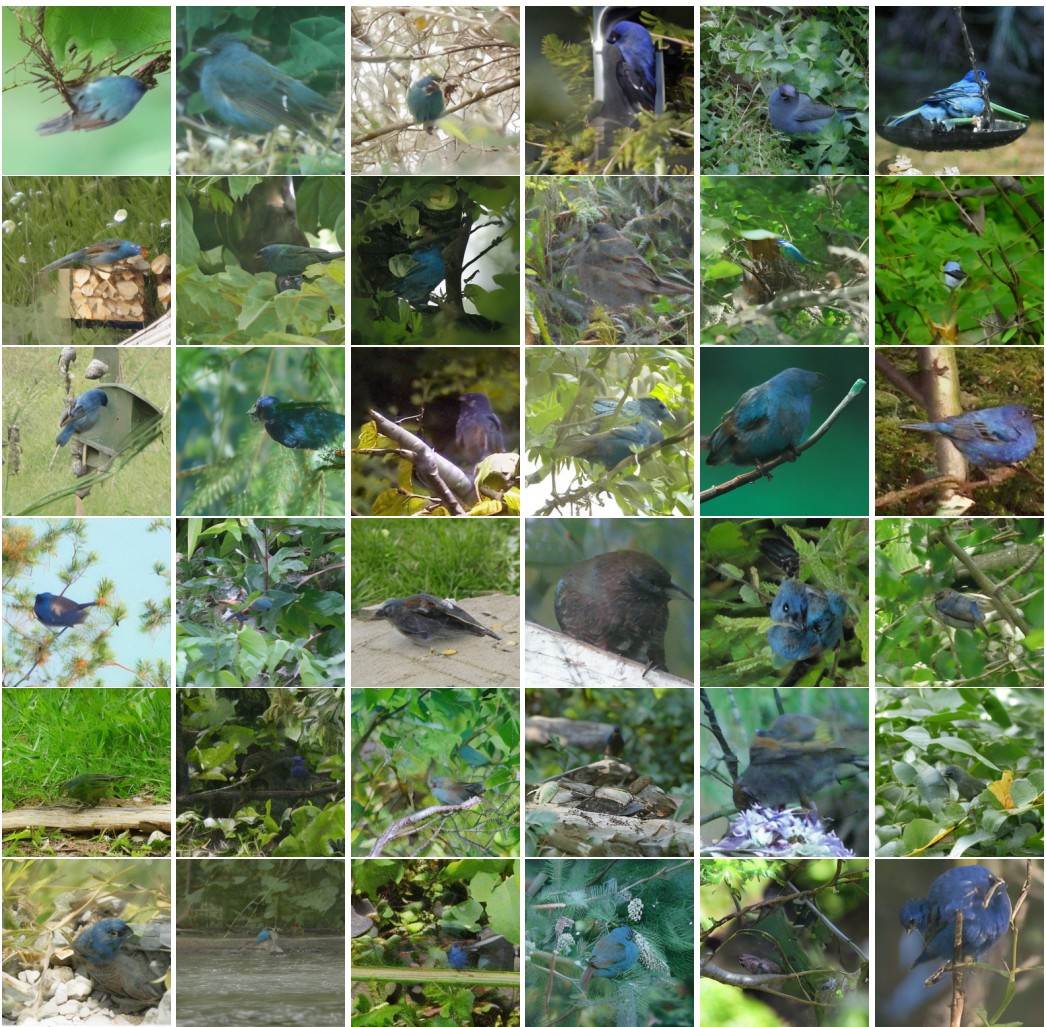

Figure 6: Selected synthetic adversarial examples on ImageNet for the class "indigo bunting". All images display a high degree of realism and are classified wrongly into various classes.

## B.5 GENERATIVE ADVERSARIAL TRANSFORMATION

In Fig. 7, we show additional examples of the GAT task. All original images are classified correctly into the ImageNet classes "golden retriever", "spider monkey", "football helmet", "jack-o'-lantern", "pickup truck", and "broccoli", while the adversarial images are classified as "cocker spaniel", "gibbon", "crash helmet", "barrel", "convertible", and "custard apple", respectively. While all adversarial images display subtle differences they do not alter the core semantics of the images and are not captured by common $\ell_p$-norms.

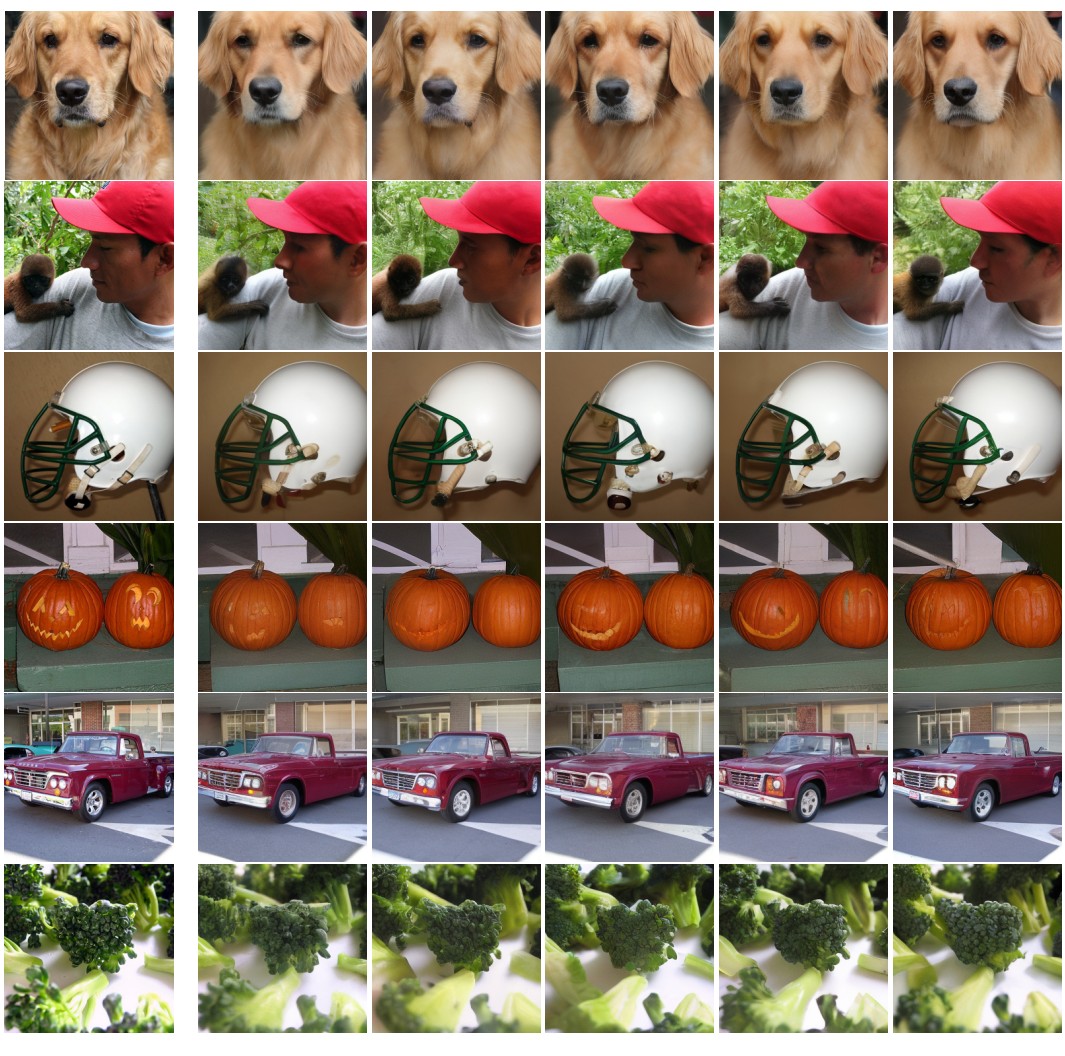

(a) Original.  (b) Adversarial Examples.

Figure 7: Selected transformed adversarial examples on ImageNet. While the adversarial examples are classified wrongly, the original images are classified correctly. All images maintain the semantics while being outside of common perturbation norms.

