# OpenReview forum: "Assessing Robustness via Score-based Adversarial Image Generation"
_ICLR.cc/2024/Conference — Submitted to ICLR 2024_

### Official Review · Reviewer_4m9f · 2023-10-30

**Soundness:** 3 good
**Presentation:** 3 good
**Contribution:** 2 fair
**Rating:** 6
**Confidence:** 3

**Summary:**

This paper proposes ScoreAG, a novel method that leverages score-based generative model to find adversarial samples. State-of-the-art methods can generate examples under the l-p-norm constraints, but such adversarial examples may be unrealistic due to the ignorance of semantic difference. To overcome these, the proposed ScoreAG introduces the diffusion models into adversarial example generation. Such score-based method is evaluated on CIFAR and TinyImagenet in its capability to synthesise from scratch, transform existing ones and purification to counter attack. Their results reveal that ScoreAG outperforms baselines in attacking effectiveness and defence competence after adversarial purification.

**Strengths:**

By introducing ScoreAG, the authors bridge the gap between adversarial attack methods and score-based generative models. This intersection is not only innovative but also timely. In contrast to the limitations seen with traditional l_p norm-based methods, ScoreAG addresses the key issue of ignoring semantic shifts during image classification tasks. This addresses a crucial gap in existing methods, ensuring that adversarial examples remain contextually relevant and semantically consistent.

Moreover, the flexibility of ScoreAG is evident in its ability to not only generate adversarial examples from pre-existing images but also to craft them from scratch. This versatility extends to its capability to purify blurry/noised samples, further emphasizing its practical utility.

By ensuring that ScoreAG aligns closely with real-world scenarios, the authors have underscored the framework's potential impact, emphasizing its relevance in practical settings. Overall, ScoreAG, unites diffusion models and adversarial attack, and offers tangible solutions for semantic-preserving image alterations.

**Weaknesses:**

Interaction between Different Tasks: The authors could explore and elucidate the interactions between different tasks such as GAS, GAT, and purification. Specifically, it would be insightful to understand the impact of applying purification post-GAS or GAT. A discussion on whether purification enhances the robustness of the adversarial examples generated by GAS or GAT would be valuable. This could provide readers with a deeper understanding of the potential synergies or trade-offs between these tasks.

Semantic Preservation: The paper could benefit from a clearer explanation and justification of the semantic requirements (Ω) used in the experiments. The methodology and criteria used to ensure semantic coherence and relevance in the generated adversarial examples should be thoroughly discussed. Consider discussing and comparing the semantic perturbations introduced by ScoreAG with those from other image-based semantic perturbation adversarial attack methods, such as the one mentioned ("SemanticAdv: Generating Adversarial Examples via Attribute-conditioned Image Editing"). Including detailed results or comparisons concerning semantic preservation could bolster the reliability and credibility of the proposed method.

Comparative Analysis in Human Study:  The human study results presented in Table 4 only focus on ScoreAG without including comparisons with the baseline methods used in other parts of the evaluation. For a more comprehensive understanding and evaluation, it would be beneficial to include a comparative analysis showcasing the performance of ScoreAG against other established methods. Consider including a discussion on the variance in naturalness or the extent of perceptual shifts observed across different methods. This could provide a more nuanced view of ScoreAG’s performance in maintaining image naturalness while generating adversarial examples.

Overall, while the paper presents an innovative approach in ScoreAG and provides a comprehensive set of experiments, there are areas where it could go deeper in comparative analysis, semantic preservation, and the discussion on task interactions.

**Questions:**

- Could you provide results and discussions comparing ScoreAG to other baseline methods in the human study section?

- Could you elaborate on the relationship between different tasks such as GAS, GAT, and purification within the ScoreAG framework? Can they be integrated into an ensemble for attack or defense?

- How does ScoreAG ensure that the generated adversarial examples maintain semantic coherence and relevance?

- Could you elaborate the perceptual shifts or changes in the naturalness of adversarial images from ScoreAG compared to other methods?

---

> ### Author Response · Authors · 2023-11-18
>
> We would like to thank the reviewer for their constructive and valuable feedback. In the following, we address their remarks and questions.
>
> **Comment:** Could you elaborate on the relationship between different tasks such as GAS, GAT, and purification within the ScoreAG framework? Can they be integrated into an ensemble for attack or defense?
> **Response:** In our framework, GAS and GAT are different forms of diffusion-based attacks, and GAP is a diffusion-based defense. The purification task GAP can be applied to defend against attacks generated by GAS and GAT. In Tab. 2, we show the performance of GAT on GAP. For convenience, we show it in the following table with the accuracy of GAS on GAP.
>
> | Method | GAT | GAS |
> | ------ | -------- | -------- |
> | GAP    | $90.74 \pm 0.67$ | $83.14\pm 1.50$     |
>
> GAP is capable of successfully defending against both attacks, GAT and GAS.
>
>
> **Comment:** How does ScoreAG ensure that the generated adversarial examples maintain semantic coherence and relevance?
> **Response:** ScoreAG ensures the semantic coherence and relevance of generated adversarial images through two key mechanisms. Firstly, we employ a class-conditional score function, i.e., $\nabla\_{\mathbf{x}\_t}\log p\_{t,y^*} (\mathbf{x}\_t)$ (see Eq. 7), ensuring that the adversarial images retain the semantic attributes of the target class $y^*$. Secondly, the guidance term, i.e., $\nabla\_{\mathbf{x}\_t}\log p\_{t,y^*}(\mathbf{x}^*,f(\mathbf{x}\_0)=\tilde{y}|\mathbf{x}\_t)$ (see Eq. 7) ensures that the generated image preserves core-semantics of $x^*$ and is classified as $\tilde{y}$. Together, these two mechanisms generate semantic-preserving adversarial images of $x^*$.
>
> **Comment:** Could you provide results and discussions comparing ScoreAG to other baseline methods in the human study section?
> **Response:** To ensure semantic preservation, we extended the evaluation of ScoreAG with a human evaluation. While we agree that these results would be interesting for other methods, human studies have significant financial costs, and thus, we think it is the responsibility of the corresponding authors to provide these. Hence, due to financial constraints, it is not feasible for us to include other baselines.
>
> **Comment:** Could you elaborate on the perceptual shifts or changes in the naturalness of adversarial images from ScoreAG compared to other methods?
> **Response:** Existing works show that diffusion models are able to capture the diversity of the training distribution [1, 2]. Furthermore, we show that our synthetic adversarial attacks have a competitive FID, indicating a high similarity between the adversarial synthetic images and real images. We can also observe a high degree of realism and diversity in our qualitative examples.
>
> We thank the reviewer again for their comments and feedback. We hope that we have satisfactorily answered the reviewer's questions. If you have any open concerns, please let us know.
>
> [1] **Bayat, Reza.** "A Study on Sample Diversity in Generative Models: GANs vs. Diffusion Models." (2023).
>
> [2] **Dhariwal, Prafulla, and Alexander Nichol.** "Diffusion models beat gans on image synthesis." Advances in neural information processing systems 34 (2021): 8780-8794.

---

> > ### Comment · Reviewer_4m9f · 2023-11-23
> >
> > Dear authors,
> >
> > Thank you for you answers and the clarification.
> >
> > I would suggest describing semantic coherence in more details in the paper, perhaps with short experiments to illustrate the correlation between the two terms and the claimed semantic preservation.
> >
> > I do not understand your answer regarding human study. Your answer states that  "it is the responsibility of the corresponding authors to provide these (financial costs)". Isn't the corresponding author among you? I would also argue that there are means to conduct simple human studies at reasonable cost, especially when it is only about comparing images.

---

> > > ### Author Response · Authors · 2023-11-23
> > >
> > > We thank the reviewer for their further suggestions.
> > >
> > > > I would suggest describing semantic coherence in more details in the paper, perhaps with short experiments to illustrate the correlation between the two terms and the claimed semantic preservation.
> > >
> > > We included several experiments and evaluations to investigate the correlation of these two terms. In Fig. 4, we show how the FID and the accuracy change when $s_y$ increases for the GAS task. Note that $s_y$ corresponds to a standard EDM generation. A higher $s_y$ leads to lower accuracy and increases the FID, indicating that the attack gets stronger but the image coherence slowly degrades. Furthermore, we observe the same in Tab. 3 for GAT. While increasing $s_y$ results in a stronger attack, increasing $s_x$ leads to a lower $\ell_2$ distance to the original image. We can observe this behavior in Fig. 3.
> > >
> > > For the camera-ready version, we plan to adapt Section 4.1 to further make the implications on the semantic coherence of the presented results more clear and thus, will discuss them in more detail.
> > >
> > >
> > > > I do not understand your answer regarding human study. Your answer states that "it is the responsibility of the corresponding authors to provide these (financial costs)". Isn't the corresponding author among you? I would also argue that there are means to conduct simple human studies at reasonable cost, especially when it is only about comparing images.
> > >
> > > Our human study cost several hundreds of euros as hundreds of images have to be classified individually by 60 human evaluators. In our case, this totaled over 32 hours of human labor for which we paid the recommended (by Prolific) hourly wage, as we believe in fair pay. Thus, each other baseline would cost again hundreds of euros. As a result, we see the responsibility to show the semantic preservation property of a method at the authors of that method (in the same way as the authors have to show the effectiveness of their proposed method in other metrics). As such an investigation is lacking in all related previous work, we see our study as significantly improving on the current evaluation standard.

---

> > > > ### Comment · Reviewer_4m9f · 2023-11-23
> > > >
> > > > I previously misunderstood what you meant by "the corresponding author". I know understand that you mena the authors of the previous work. Thanks for the clarification.

---

> > > > > ### Comment · Reviewer_4m9f · 2023-11-23
> > > > >
> > > > > Regarding the human study, I appreciate the fact that they can be costly. A reasonable compromise is to conduct a smaller study on the baselines. This would provide supporting evidence that the score-based approach preserves semantics better than the compared baselines.

---

### Official Review · Reviewer_rCh2 · 2023-10-31

**Soundness:** 3 good
**Presentation:** 3 good
**Contribution:** 2 fair
**Rating:** 5
**Confidence:** 5

**Summary:**

This paper proposes ScoreAG, a framework for generating unrestricted adversarial examples using score-based generative models with diffusion guidance. The key idea is to leverage the generative capabilities of these models to synthesize new adversarial images from scratch (GAS), transform existing images into adversarial ones (GAT), and purify images to enhance classifier robustness (GAP).

**Strengths:**

The paper is well-written and the proposed method is novel. Generating semantic-preserving adversarial examples beyond standard threat models is an important research direction, and the use of score-based diffusion models is a promising approach. The experiments are extensive, comparing ScoreAG to several state-of-the-art attacks and defenses across multiple datasets. The results demonstrate ScoreAG's effectiveness in crafting unrestricted adversarial examples.

**Weaknesses:**

- The notion of "unrestricted" adversarial examples needs more discussion. While ScoreAG does not use an explicit lp norm, the samples are still constrained to the manifold learned by the generative model. Analyzing the diversity/range of examples is important.

- More analysis is needed on why ScoreAG outperforms the other diffusion-based attack DiffAttack. The reasons are not fully clear.

- The lack of certified or provable robustness guarantees for the purified models is a limitation. Evaluating security empirically is difficult.

- The human study provides useful insights but is limited in scope. Expanding this to quantify semantic similarity and diversity of examples would strengthen the evaluation.

- Lacking important references like "Content-based Unrestricted Adversarial Attack"

**Questions:**

see weaknesses

---

> ### Author Response · Authors · 2023-11-18
>
> We want to thank the reviewer for their feedback and questions. In the following, we address their comments.
>
> **Comment:** Notion of "unrestricted" needs more discussion. Samples are still constrained to the manifold learned by the generative model. Analyzing the diversity/range of examples is important.
> **Response:**
> We use the term "unrestricted" to emphasize that the adversarial attacks generated by ScoreAG are not bounded by any common $\ell_p$-norm. Thereby, we maintain a consistent terminology with related work [1, 2]. As correctly pointed out by the reviewer, it is important to clarify that while ScoreAG transcends traditional $\ell_p$-norm constraints, it still operates within the learned manifold of the generative model. This approach aligns with our goal of creating adversarial examples that are semantically meaningful and realistic rather than strictly adhering to normative limitations. We clarified this in the updated manuscript.
>
> **Comment:** Why does ScoreAG outperform DiffAttack?
> **Response:** While the ScoreAG and DiffAttack have many differences, we assume ScoreAG's key advantage is the ability to go through the whole latent space by utilizing diffusion guidance, while DiffAttack is limited to the last few steps. To verify this empirically, we compare how the performance of ScoreAG changes in the GAT setting when only denoising the last few steps.
>
>
>
> | Full reverse | 5 reverse steps | 10 reverse steps | 20 reverse steps |
> | -------- | -------- | -------- | ---- |
> | 0.0     | 0.28     | 0.21     | 0.09 |
>
> The table shows the accuracy after applying GAT on the first 100 images of Cifar-10. As expected, 5 and 10 reverse steps, i.e., the values used by DiffAttack, are not sufficient for a successful attack.
>
>
> **Comment:** Lack of certifiable robustness is a limitation.
> **Response:** We now calculated certifiable robustness using randomized smoothing. See the general comment for more information.
>
> **Comment:** The human study is limited in scope. Expanding this to quantify semantic similarity and diversity of examples would strengthen the evaluation.
> **Response:** We agree with the reviewer that quantifying semantic similarity and diversity of the adversarial examples are important but not trivial as well. In our human study, we implicitly quantify similarity by capturing the semantic preservation through the human accuracy. For the GAT task, the human accuracy reaches 94%, implying semantic preservation of the images. The diversity, however, is correlated with the capabilities of the generative model and the dataset itself. We quantify the diversity of the generative model using the FID score in Fig.4 (left).
>
> **Comment:** Missing references to unrestricted methods.
> **Response:** We thank the reviewer for that reference. We added it in the updated manuscript.
>
> We thank the reviewer again for their comments and feedback. We hope that we have satisfactorily answered the reviewer's comments. If you have any open concerns, please let us know.
>
> [1] **Chen, Jianqi, Hao Chen, Keyan Chen, Yilan Zhang, Zhengxia Zou, and Zhenwei Shi.** "Diffusion Models for Imperceptible and Transferable Adversarial Attack." arXiv preprint arXiv:2305.08192 (2023).
>
> [2] **Song, Yang, Rui Shu, Nate Kushman, and Stefano Ermon.** "Constructing unrestricted adversarial examples with generative models." Advances in Neural Information Processing Systems 31 (2018).

---

> > ### Comment · Reviewer_rCh2 · 2023-11-23
> > **Additional Concern**
> >
> > Although the authors answered my questions in the response, I found a recent NeurIPS【1】 paper that is extremely similar to this paper, with the main difference being the use of a different diffusion model in the generation of adversarial examples. Due to the above issues, I believe that the technical novelty of this paper still needs to be further improved, so I will maintain the current score for now.
> >
> > 【1】Content-based Unrestricted Adversarial Attack
> > https://neurips.cc/virtual/2023/poster/70854

---

> ### Author Response · Authors · 2023-11-23
>
> We thank the reviewer for their reply and the reference.
>
> The mentioned work by Chen et al. (2023) was only recently published at NeurIPS 2023 and is thereby categorized as concurrent under ICLR guidelines:
>
> > We consider papers contemporaneous if they are published (available in online proceedings) within the last four months. That means, since our full paper deadline is September 28, if a paper was published (i.e., at a peer-reviewed venue) on or after May 28, 2023, authors are not required to compare their own work to that paper.
>
> Furthermore, we cannot make an empirical comparison as the authors have not published their code yet. Nevertheless, we see several differences between their and our work. Most importantly,
> - They perform a forward DDIM process to find the latent representation of an intermediate step, while we start from noise and do not require a latent mapping of the images.
> - While we iteratively denoise our adversarial image using guided diffusion, Chen et al. (2023) refine their adversarial perturbation staying in the same latent step.
> - Finally, ScoreAG leverages the whole diffusion process and latent space.
>
> We will include a detailed discussion about the differences in the related work of the final manuscript.

---

### Official Review · Reviewer_dx4f · 2023-10-31

**Soundness:** 2 fair
**Presentation:** 2 fair
**Contribution:** 2 fair
**Rating:** 3
**Confidence:** 4

**Summary:**

The paper introduces a novel framework called Score-Based Adversarial Generation (ScoreAG) for generating adversarial examples beyond the constraints of $\ell_p$-norms. The authors leverage advancements in score-based generative models to synthesize semantic-preserving adversarial examples, transform existing images into adversarial ones, and purify images to achieve adversarial robustness. They conduct an empirical evaluation on CIFAR-10, CIFAR-100, and TinyImageNet datasets to demonstrate the effectiveness of ScoreAG.

**Strengths:**

The main contribution of this paper is the introduction of Score-Based Adversarial Generation (ScoreAG) and the following three uses of it: Synthesis (Generative Adversarial Synthesis, GAS), transformation (Generative Adversarial Transformation, GAT), and purification (Generative Adversarial Purification, GAP). This paper is well-structured and the method is described clearly.

**Weaknesses:**

This work is subject to several weaknesses that need to be addressed.

1. Although GAS aims to generate the adversarial example from scratch that would be misclassified by the classifier while preserving the semantics of the truth class, it is disappointing to see that even human fails to classify the adversarial examples. As shown in Table 4, Human accuracy on the adversarial synthetic images is only 70%, this results do not support the argument that it preserves the semantics of a certain class.

2. The paper argues that GAS provides a more *comprehensive robustness assessment*, but lacks quantitative results to support this assertion.

3. A comparison with similar work [1] should be provided, along with an explanation of the differences in approach and findings.

4. GAT transforms clean images into adversarial examples, but the comparison is limited to DiffAttack. Including other unrestricted attacks [2, 3, 4] in the comparison would provide a more thorough assessment. Additionally, reporting the mean and standard deviation for the results would enhance statistical robustness.

5. It is unclear what the "ScoreAG" column represents in Table 2. If it indicates GAP, it is unclear what the given adversarial image ($\mathbf{x}_\text{ADV}$) is used to purify. If it is GAT, isn’t the higher robust accuracy representing the weaker attack efficacy? Further clarifications are needed to understand the evaluation

6. The robustness results (Table 2) should be compared with more baselines, including both attacks and defenses, such as DiffAttack and [1, 2, 3, 4]. Additionally, results should be reported for all datasets, not just CIFAR-10, to provide a comprehensive view.

7. The claim that GAP further enhances the adversarial robustness of the model is not sound. Since, generally, adversarial robustness is to evaluate the model’s accuracy under perturbations, the purification seems to serve merely as a defense approach. Therefore, it should be compared with other adversarial defense methods, such as [5, 6].

8. In Section 4.2, the author argues that the $\ell_p$-bounded methods display noticeable noisy fragments. However, I think it is because of cherry-picking. Also, I do not understand where the statement “*the removal of a small fish — which prove to be important classification cues*” comes from.

9. The authors did not submit the code for review. If the authors want to avoid releasing publicly before acceptance, submitting to AC privately is a viable option.

10. The practical application of the proposed approach should be discussed. How can it be used in real-world scenarios? How is it different from other approaches in practice?

11. The visualization on ImageNet seems to have high quality, it would be great to report the experimental results on it to demonstrate the generalizability of the proposed approach.

**[After Rebuttal]**

The reviewer thanks the authors for the rebuttal. However, the reviewer thinks there are still many concerns that remain unresolved. For example, the following two statements seem to be in conflict: "*GAT aims to achieve a low robust accuracy ...*" and "*GAT achieves a slightly better accuracy than APGD and APGDT in many cases, implying a more comprehensive robustness assessment*." Why does a higher accuracy represent a more comprehensive robustness assessment? Also, the computational cost of GAT is very high, and its performance does not justify it. In addition, I encourage authors to provide GAT and GAP results on TinyImageNet and include other adversarial training baselines [2, 3] in Table 2.

>   [1] Chen et al. AdvDiffuser: Natural Adversarial Example Synthesis with Diffusion Models (ICCV 2023)
>
>   [2] Hsiung et al. Towards Compositional Adversarial Robustness: Generalizing Adversarial Training to Composite Semantic Perturbations (CVPR 2023)
>
>   [3] Laidlaw et al. Perceptual Adversarial Robustness: Defense Against Unseen Threat Models (ICLR 2021)
>
>   [4] Bhattad et al. Unrestricted adversarial examples via semantic manipulation (ICLR 2020)
>
>   [5] Frosio and Kautz. The Best Defense is a Good Offense: Adversarial Augmentation against Adversarial Attacks (CVPR 2023)
>
>   [6] Cohen et al. Certified Adversarial Robustness via Randomized Smoothing (ICML 2019)

**Questions:**

Please refer to the weakness. In addition, please also address the following questions.

1. Table 1 only considers one model architecture (WRN-28-10), please provide more models to verify the efficacy of the proposed method.
2. What is the computational cost of ScoreAG compared to other methods?

---

> ### Author Response · Authors · 2023-11-18
>
> We would like to thank the reviewer for their valuable feedback. In the following, we address their raised concerns.
>
> **Comment:** Human accuracy on the adversarial synthetic images is only 70%.
> **Response:** The generative adversarial transformations (GAP) achieve an almost perfect human accuracy of 94%, equal to the accuracy on the non-adversarial synthetic images. This indicates that the adversarial attack does not degrade the semantic coherence for humans. We respectfully disagree with the reviewer that humans fail to classify the synthetic adversarial images. This is a novel task, and the human accuracy is 70%, which is 60% higher than guessing, implying a success. We expect future work to further improve these results with the development of better generative models.
>
> **Comment:** Lack of quantitative results for the claim of a *more comprehensive robustness assessment*.
> **Response:** GAS and GAT conceptually provide a more comprehensive robustness assessment than $\ell_p$-bounded adversarial attacks, as (i) all semantic-preserving adversarial examples within the $\ell_p$-balls are part of GAS and GAT given a reasonably trained generative model, and (ii) further incorporate semantic-preserving adversarial examples not captured by common $\ell_p$-threat models. Therefore, our claim of a more comprehensive robustness assessment stems from the inherent capability of GAS and GAT to cover a wider spectrum of adversarial examples, including those restricted by $\ell_p$-norm constraints. We want to note that we do not necessarily see GAS and GAT as a replacement for traditional attack methods but as complementary to explore semantic-preserving areas that are not included in common $\ell_p$-balls.
>
> **Comment:** Comparison with similar work should be provided.
> **Response:** We would like to thank the reviewer for the interesting reference. However, as the referenced paper was only recently published at ICCV2023, we would like to remind the reviewer that this work is categorized as concurrent under ICLR guidelines:
> > We consider papers contemporaneous if they are published (available in online proceedings) within the last four months. That means, since our full paper deadline is September 28, if a paper was published (i.e., at a peer-reviewed venue) on or after May 28, 2023, authors are not required to compare their own work to that paper.
>
> However, we include a discussion of it in the related work section. Unfortunately, we are not able to include it in our experiments, as the code is not publicly available yet.
>
> **Comment:** Unclear what Table 2 represents.
> **Response:** The column labeled "ScoreAG" represents the attack GAT, while the corresponding row represents the purification GAP. We added this information to the table. While GAT aims to achieve a low robust accuracy, GAP aims to achieve a high accuracy. GAT achieves a slightly better accuracy than APGD and APGDT in many cases, implying a more comprehensive robustness assessment.

---

> ### Author Response · Authors · 2023-11-18
>
> **Comment:** Table 1 and Table 2 should contain more baselines, classifiers, and standard deviations.
> **Response:** We added standard deviations to Table 1 and Table 2. Furthermore, we now added two more attacks to Table 1, resulting in a total of 13 (excluding ScoreAG) evaluated attacks. Additionally, we also added four more classifiers evaluated on ScoreAG (GAT) to further demonstrate its efficacy. The following table shows the adversarial accuracy of GAT in % using the same hyperparameters as for the WRN-28-10 architecture.
>
> | Classifier | Cifar-10 | Cifar-100 |
> | -------- | -------- | -------- |
> | ResNet-20 | 0.01 | 0.10     |
> | ResNet-56 | 0.03 | 0.13     |
> | VGG-19 | 0.52 |  1.94  |
> | RepVGG-A2 | 0.04 | 0.26 |
>
> Note that the models are obtained from PyTorch Hub. Therefore, we cannot report standard deviations. As we can observe, ScoreAG successfully generates adversarial attacks using different classifiers. We added these results to the updated manuscript.
>
> In Table 2, we already compare ScoreAG to six state-of-the-art defenses and purification methods. However, we added an additional comparison to evaluate the certified robustness of GAP and randomized smoothing (see general comment). Note that the purification experiments require unconditional generative models as the ground-truth class labels are unknown. Therefore, our purification experiments are focused on the CIFAR-10 dataset, given the availability of unconditional pre-trained EDM models and extensive prior work [1, 2].
>
>
> **Comment:** Authors claim that $\ell_p$-bounded methods display noticeable noisy fragments. Was it cherry-picking? Also, where does the *the removal of a small fish* statement come from?
> **Response:** The images were not cherry-picked, and we observed the same noisy fragments across all samples. [We uploaded more **randomly chosen** example images for APGD, APGDT, Square, and ScoreAG to Figshare.](https://figshare.com/s/117b453e5289311df4c7) The importance of the removal of the small fish refers to Fig. 1a and 1b, meaning the observation that many of the images generated by ScoreAG remove or modify the fish next to the shark (representing the image class) in the image.
>
> **Comment:** Practical application of ScoreAG.
> **Response:** ScoreAG only requires a pre-trained score-based generative model and the gradients of a classifier. Therefore, it can be applied in any scenario where other attacks are currently used. Furthermore, traditional $\ell_p$-threat models often do not capture important semantics-preserving corruption relevant to real-world applications [3, 4, 5]. As ScoreAG effectively overcomes this limitation, it can be highly interesting for evaluating robustness in real-world scenarios.
>
> **Comment:** Why are there no experimental results on ImageNet?
> **Response:** Many methods in our comparison already reach a near-perfect success rate on TinyImageNet, which is a subset of ImageNet. As ImageNet contains five times as many labels and a higher resolution, it is an easier target. However, we included ImageNet in our qualitative evaluation due to its high resolution.
>
> **Comment:** What is the Runtime of ScoreAG?
> **Response:** We added a runtime comparison to other baselines and the EDM generative model itself. See the general comment for more information.
>
> We again thank the reviewer for their comments and hope that we satisfactorily addressed them. If not, we are happy to address any remaining concerns.
>
> [1] **Wang, Zekai, Tianyu Pang, Chao Du, Min Lin, Weiwei Liu, and Shuicheng Yan.** "Better diffusion models further improve adversarial training." arXiv preprint arXiv:2302.04638 (2023).
>
> [2] **Karras, Tero, Miika Aittala, Timo Aila, and Samuli Laine.** "Elucidating the design space of diffusion-based generative models." Advances in Neural Information Processing Systems 35 (2022): 26565-26577.
>
> [3] **Eykholt, Kevin, Ivan Evtimov, Earlence Fernandes, Bo Li, Amir Rahmati, Chaowei Xiao, Atul Prakash, Tadayoshi Kohno, and Dawn Song.** "Robust physical-world attacks on deep learning visual classification." In Proceedings of the IEEE conference on computer vision and pattern recognition, pp. 1625-1634. 2018.
>
> [4] **Kar, Oğuzhan Fatih, Teresa Yeo, Andrei Atanov, and Amir Zamir.** "3d common corruptions and data augmentation." In Proceedings of the IEEE/CVF Conference on Computer Vision and Pattern Recognition, pp. 18963-18974. 2022.
>
> [5] **Hendrycks, Dan, Nicholas Carlini, John Schulman, and Jacob Steinhardt.** "Unsolved problems in ml safety." arXiv preprint arXiv:2109.13916 (2021).

---

### Official Review · Reviewer_3Sog · 2023-11-01

**Soundness:** 2 fair
**Presentation:** 3 good
**Contribution:** 3 good
**Rating:** 5
**Confidence:** 4

**Summary:**

This work presents a framework, ScoreAG, designed for adversarial example generation beyond \mathcal{l}_{p}-norm constraints. By integrating Score-based Generative Modelling with Diffusion Guidance techniques, ScoreAG effectively addresses the limitations of conventional adversarial generation approaches. By applying various conditional guidance terms in the reverse-time SDE process, ScoreAG can 1) synthesize adversarial images (GAS), 2) transform adversarial images (GAT), and 3) purify adversarial images (GAP). Experimental results show that ScoreAG can outperform leading benchmarks in adversarial attack and defense, generating adversarial examples while preserving their original semantic meaning.

**Strengths:**

1. The paper is well-organized and easy to follow.

2. Conducting adversarial attacks using semantic-bounded examples offers an interesting viewpoint on robustness evaluation.

3. As demonstrated in the experimental section, the proposed framework attains promising performance across three tasks.

**Weaknesses:**

1. Some formulas in this paper were derived incorrectly, such as equation 7 on page 4. The authors should check the methodology section to ensure correctness.

2. The authors compare the attack effectiveness of GAT with other benchmark methods in Section 4.1. Given that generation efficiency is also an essential factor in evaluating the effectiveness of adversarial attacks, it is recommended that the authors add a discussion on the efficiency of adversarial sample generation to this part of the experiment.

3. In Table 2, the evaluation results are displayed for Cifar-10. To ensure the applicability of the proposed method, evaluations on datasets of more categories or higher resolution, such as Cifar-100 and TinyImageNet, are advisable. Experimental results in Table 2 also compare the efficacy of various adversarial training and purification methods. However, the distinct architectures used by these methods raise concerns about the fairness of this comparison.

4. This paper lacks details of the experimental implementation environments, which limits the reproducibility of the proposed method.

5. As pointed out in Section 4.3, some misclassification cases are due to the low resolution of Cifar-10’s images, indicating that employing higher-resolution datasets for testing would have been more appropriate.

**Questions:**

1. How time-efficient is the method you’ve proposed?

2. With the distinct architectures used by methods presented in Table 2, can we still view this comparison as fair?

3. Why not use images from higher-resolution datasets for testing in Section 4.3?

---

> ### Author Response · Authors · 2023-11-18
>
> We want to thank the reviewer for their constructive and valuable feedback. In the following, we address their remarks and questions.
>
> **Comment:** Equation 7 is incorrect.
> **Response:** We corrected the typo in the new manuscript.
>
> **Comment:** Table 2 does not include Cifar-100 and TinyImagenet. Architectures are different for various methods; is the comparison fair?
> **Response:** Our method uses the WRN-28-10, a widely used architecture for adversarial attacks and robustness. To make a fair comparison, we selected the best models using the same architecture and the best models overall, which use the WRN-70-16 architecture. Therefore, this is a biased comparison in favor of the WRN-70-16 architectures and the baselines. Note that the purification experiments require unconditional generative models as the ground-truth class labels are unknown. Therefore, our purification experiments are focused on the CIFAR-10 dataset, given the availability of unconditional pre-trained EDM models and extensive prior work [1, 2].
>
> **Comment:** Why not use higher-resolution datasets for the human study?
> **Response:** While selecting the dataset for the human trial, we prioritized maintaining the participant's accessibility, attention, and engagement. Moreover, datasets such as ImageNet contain classes that require specialized domain knowledge, e.g., "barn spider" and "garden spider". Although selecting a subset of ImageNet classes is an option, we ultimately decided against it to avoid introducing selection bias, which could skew the study's outcomes. This led us to select a dataset with a limited number of classes.
>
> **Edit (20.11):**
>
> **Comment:** Lack of details of the experimental implementation environment.
> **Response:** We added a reproducibility statement and uploaded the code of ScoreAG. See the general comment for more information.
>
> **Comment:** What is the runtime of ScoreAG?
> **Response:** We added a runtime comparison to the appendix. For more details, see the general comment.
>
> We thank the reviewer again for their comments and feedback. We hope that we have satisfactorily answered the reviewer's questions. If you have any open concerns, please let us know.
>
> [1] **Wang, Zekai, Tianyu Pang, Chao Du, Min Lin, Weiwei Liu, and Shuicheng Yan.** "Better diffusion models further improve adversarial training." arXiv preprint arXiv:2302.04638 (2023).
>
> [2] **Karras, Tero, Miika Aittala, Timo Aila, and Samuli Laine.** "Elucidating the design space of diffusion-based generative models." Advances in Neural Information Processing Systems 35 (2022): 26565-26577.

---

> > ### Comment · Reviewer_3Sog · 2023-11-23
> >
> > Thank you for your response. Most of my concerns have been addressed. However, for the human study, even a small-scale experiment on a higher-resolution dataset would significantly strenghthen the soundness of this paper.

---

### Author Response · Authors · 2023-11-18

We thank the reviewers for their constructive feedback and for noting the **novelty of our work** (dx4f, rCh2, 4m9f), the **clear and well-structured writing** (dx4f, 3Sog, rCh2), that we **fill a crucial gap** in existing methods (4m9f), and that we effectively **address the limitations of conventional adversarial generation approaches** (3Sog).

Individual responses to reviewers are available under each review. In this general response, we seek to address reoccurring  comments:

**Certified accuracy. (#rCh2, #dx4f)**
We demonstrate that certifying the robustness of the purified model is possible using randomized smoothing. By integrating the purification into the classification process, we create a robust classifier whose certified accuracy can be evaluated. To illustrate this, we present the certified accuracy (in %) for the Cifar-10 dataset by employing randomized smoothing:


| Method | $\sigma=0.25$ | $\sigma=0.5$ |
| -------- | -------- | -------- |
| Base Classifier |   8.43   |   10.00   |
| Base Classifier + GAP | 71.07     | 31.85     |

The table compares the certified accuracy of the base classifier against that of the base classifier enhanced with GAP (Gradient Adversarial Purification), using $N=100$ samples and different noise levels $\sigma$. The significant increase in certified accuracy underscores its efficacy in enhancing robustness.


**Runtime of ScoreAG (#3sog, #dx4f)**
As GAT is capable of generating perturbations beyond the typical $\ell_p$-ball threat model, it is most appropriate to compare it to other generative model-based attacks, such as DiffAttack, which is more computationally intensive than GAT. The following table shows the average runtime for a batch of 16 images on a 1080Ti for several baselines and the generative model EDM. We were not able to run DiffAttack even with a batch size of one on this GPU.


|FGSM | DIFGSM | SINIFGSM | Square | FAB | APGD | APGDT | OnePixel | LPA | PPGD |DiffAttack | GAT | EDM |
| --- | ------ | -------- | ------ | --- | ---- | ----- |--------- |---- | ---- | ------- | ----- | ---
|0.01 | 0.27 | 1.73 | 67.43 | 7.03 | 0.44 | 0.48 | 2.62 | 41.24 | 2.06 | OOM | 246.06 | 79.47 |

In comparison with PGD, GAT has a higher computational overhead.

**Implementation details and code. (#3sog, #dx4f)**
To improve reproducibility, we included a reproducibility statement in the updated manuscript detailing implementation details. Furthermore, we included an overview of ScoreAG's hyperparameters in the Appendix and shared the code in a private comment and will make it public upon acceptance.

We again thank all the reviewers and are confident that we improved the manuscript based on these comments. We highlighted important changes in blue and are happy to address any remaining concerns.

---

### Author Response · Authors · 2023-11-18
**Code for Reproducibility**

Dear reviewers and AC(s),

In the following link, we share the code used to compute the GAT results in our paper.

https://figshare.com/s/5107b4693cd4589ee7bf

Best regards,
Authors

---

### Meta-Review · Area_Chair_8bid · 2023-12-12

**Metareview:**

The rebuttal addressed some concerns, but in the end, all reviewers agree that the paper is not ready for publication, so should not be accepted.

**Justification For Why Not Higher Score:**

No reviewer suggests accepting this paper, so there is no reason for choosing a better rating. The main weaknesses include lack of sufficient baselines, small scale of the human study, amount of perturbation on the image, and high computation costs.

**Justification For Why Not Lower Score:**

N/A

---

### Decision · Program_Chairs · 2024-01-16

Reject